# Investigating microscale patchiness of motile microbes under turbulence in a simulated convective mixed layer

**Alexander Kier Christensen**[1]*, **Matthew D. Piggott**[2], **Erik van Sebille**[3], **Maarten van Reeuwijk**[4], **Samraat Pawar**[1]

**1** Department of Life Sciences, Imperial College London, London, United Kingdom, **2** Department of Earth Science and Engineering, Imperial College London, London, United Kingdom, **3** Institute of Marine and Atmospheric Research, Utrecht University, Utrecht, The Netherlands, **4** Department of Civil and Environmental Engineering, Imperial College London, London, United Kingdom

\* a.christensen17@imperial.ac.uk

**Data Availability Statement:** The outputs of the microbial individual-based model simulations are hosted publicly with the Open Science Framework at link https://doi.org/10.17605/OSF.IO/72YNH

## Abstract

Microbes play a primary role in aquatic ecosystems and biogeochemical cycles. Spatial patchiness is a critical factor underlying these activities, influencing biological productivity, nutrient cycling and dynamics across trophic levels. Incorporating spatial dynamics into microbial models is a long-standing challenge, particularly where small-scale turbulence is involved. Here, we combine a fully 3D direct numerical simulation of convective mixed layer turbulence, with an individual-based microbial model to test the key hypothesis that the coupling of gyrotactic motility and turbulence drives intense microscale patchiness. The fluid model simulates turbulent convection caused by heat loss through the fluid surface, for example during the night, during autumnal or winter cooling or during a cold-air outbreak. We find that under such conditions, turbulence-driven patchiness is depth-structured and requires high motility: Near the fluid surface, intense convective turbulence overpowers motility, homogenising motile and non-motile microbes approximately equally. At greater depth, in conditions analogous to a thermocline, highly motile microbes can be over twice as patch-concentrated as non-motile microbes, and can substantially amplify their swimming velocity by efficiently exploiting fast-moving packets of fluid. Our results substantiate the predictions of earlier studies, and demonstrate that turbulence-driven patchiness is not a ubiquitous consequence of motility but rather a delicate balance of motility and turbulent intensity.

## Author summary

Understanding how spatial patchiness in aquatic microbes develops at different scales is crucial for understanding their interactions, their population dynamics, and their role in the wider ecosystem. Patchiness in microbial populations at very small scales is hard to measure or model, particularly where turbulence is involved, and patch formation mechanisms remain poorly understood. In this study, we simulated both swimming and passive

OceanParcels code is publicly hosted at https://oceanparcels.org/index.html Our microbe individual-based model ran on top of OceanParcels v2.1.4 (Github commit hash ceb0e42b69046bc6dcb6540ac08268db1855f1e2). All original code pertaining to our microbe individual-based model is hosted on Github at: https://github.com/christensen5/turbulence-patchiness-sims/tree/turbulence-patchiness-paper.

**Funding:** AKC was supported by the UK Natural Environment Research Council (NERC) through the Quantitative Methods in Ecology and Evolution (QMEE) CDT, under grant NE/P012345/1. SP was supported by Leverhulme Research Fellowship (RF-2020-653\2). MP acknowledges support from the UK Engineering and Physical Sciences Research Council (EPSRC) under grant EP/R029423/1. MvR acknowledges support from the UK turbulence consortium for computational resources (EPSRC grant EP/R029326/1). EvS was supported by the Dutch Research Council (NWO) through the ENW-Klein research programme with project number OCENW.KLEIN.085 The funders had no role in study design, data collection and analysis, decision to publish, or preparation of the manuscript.

**Competing interests:** The authors have declared that no competing interests exist.

microbes in a realistic model of small-scale turbulence at an unprecedented resolution. We find that patchiness is triggered far below the surface and only among highly agile swimmers. This demonstrates that microbial patchiness can develop at sub-metre scales within realistic turbulent flows, albeit under restricted conditions. Our results predict that that strong turbulence near the surface of large water bodies (such as oceans and lakes) generated by night-time or cold weather conditions inhibits patch formation, and that patchiness is triggered primarily in deeper waters near the thermocline—a region of transition between warm surface waters and cooler waters at greater depth. Our findings highlight the sensitive balance of conditions needed to trigger patchiness in realistic flows, and demonstrate how small differences in individual behaviour can produce substantially different outcomes in the population as a whole.

## Introduction

Life on Earth is predominantly microbial [1], with microbes responsible for the majority of the metabolic activity that maintains the basic habitability of the planet [2, 3]. Aquatic microbes account for over 50% of global primary productivity [4], and are also largely responsible for the decomposition of organic matter and recycling of nutrients [5–8]. These functions are driven by complex interactions between and among microbial individuals and their environment. Precisely because they comprise so many connected components, accurately modelling these complex interactions and understanding how they play out across spatial and temporal scales, remains a major empirical and theoretical challenge [9].

Spatial heterogeneity, or "patchiness", is a critical component of aquatic microbial communities. Patchiness at large scales ($\gtrsim 0.5$ km) has been documented since at least the 1930s [10, 11], and though the traditional assumption that turbulence would homogenise microbe distributions at smaller scales [12–14] held for much longer, patchiness is increasingly understood to be common across spatial scales from millimetres to 100s of kilometers [15–18].

Patchiness matters because it can have both negative and positive effects on microbial populations, leading to wider ecosystem-level consequences. For example, microbial growth rates are higher within patches formed in nutrient hotspots or dissolved organic matter (DOM) plumes [19, 20]. On the other hand, microbes experience increased mortality when planktonic predators leverage their own motility and sensory abilities to exploit patches [21], or because of increased viral transmission rates in patch-dwelling microbes [22, 23]. Furthermore, patchiness at the smallest ($< 1$ m) scales has its own particular suite of consequences, intensifying competition for nutrients within microbe patches [9], colonising disproportionately high-growth micro-habitats [20] and establishing a basis for the formation of patches of other organisms of higher trophic levels [24]. The effects of patchiness on microbial populations can ultimately impact the dynamics of the wider ecosystem. For example, temporal or spatial separation of phytoplankton and zooplankton patches can increase primary productivity several-fold relative to a homogeneous environment [25], the aggregation of diatom detritus can increase bacterial species richness and abundance [26], and strong patchiness in competing plankton species has even been proposed as an explanation for Hutchison's long standing "paradox of the plankton" [27, 28].

Spatial dynamics and microbial patchiness are thus critical to understanding aquatic ecosystems. However, measuring and modelling their influence is difficult due to the variety and complexity of flow regimes present in nature. Turbulent flows are a particularly challenging area of research; naturally-occurring turbulence generates vortices and fluid velocity

fluctuations down as far as the sub-millimetre scale [29], rendering fully-resolved simulations computationally expensive, and necessitating state-of-the-art technologies such as high-resolution fluorometry [17, 30] or underwater imaging [31] for accurate experimental measurements. Notwithstanding these difficulties, uncovering the mechanisms responsible for widespread small-scale spatial patchiness remains essential, and in recent years, research at the intersection of ecology and fluid dynamics has begun to present candidate mechanisms [32–34].

In this paper, we address the hypothesis that gyrotactic microbial motility interacts with microscale turbulence to trigger intense patchiness, increasing local microbe concentrations by an order of magnitude or more [32, 35–37]. The aggregation effect has been hypothesised to be driven by a coupling between fluid shear (which acts to overturn or 'disorient' gyrotactic swimmers) and motility (by which the swimmers attempt to re-orient towards the vertical); when a suitable balance is achieved between the overturning effect of shear, and the swimmers' inherent stabilising torque, intense patchiness results [32]. So far however, this hypothesis has been tested only in simplified or idealised turbulence regimes that are well-suited to mathematical analysis and simulation, but do not accurately reflect the turbulent environment that microbes experience in, for example, lakes or oceans [37]. Experiments in steady vortices, for example, fail to capture the complexity of real turbulence, which comprises many unsteady vortices of different sizes emerging and dissipating constantly. More complex simulations of microbial patchiness in statistically steady-state isotropic turbulence (e.g. [38]) are a more accurate approximation of real world conditions, but it remains the case that turbulence in, for example, the ocean mixed layer is not steady-state or isotropic. Although more recent studies have begun to simulate microbial dynamics in anisotropic and nonhomogeneous flows [39], and in doing so have shed light on the relative influences of, for example, small-scale turbulence vs large-scale advection on microbial dynamics, it remains an open question to what extent previous findings regarding patchiness apply to complex turbulent regimes that mimic real-world conditions.

To address this, we developed a fully 3D microbe individual-based model (IBM) resolved at the sub-metre scale and coupled with a direct numerical simulation (DNS) of convective turbulence in a scaled-down mixed layer, and simulated microbial spatial dynamics at a range of depths. The use of an IBM allows us to resolve fine-scale spatial differences in fluid velocity and its coupling with individual motility to recover the emergent spatial dynamics of microbes and the formation of microscale patches. The fluid DNS reproduces density gradients that drive spatial fluctuations in fluid buoyancy, to produce a depth-varying anisotropic turbulent flow. Seeding the flow with both non-motile (passive) and motile (gyrotactic) virtual microbes, we tested whether the proposed mechanism of turbulence-driven patchiness is realisable in flow conditions comparable to those that a microbe would experience in a convective mixed layer, such as occurs in oceans or lakes during the night, during autumnal or winter cooling, or during a cold-air outbreak [40, 41]. We find that under such conditions, turbulence-driven patchiness is highly dependent on motility and is strongly depth-structured. In doing so we provide evidence in support of the predictions of earlier studies involving simpler models of turbulence, and demonstrate the complexity of spatial dynamics precipitated by the interaction of realistic microscale turbulence and individual motility.

## Results

### Depth-structure of the simulated flow

Our model seeks to emulate patchiness emerging in real time from the complex interaction between microbial motility and physical characteristics of the turbulent flow. The fluid

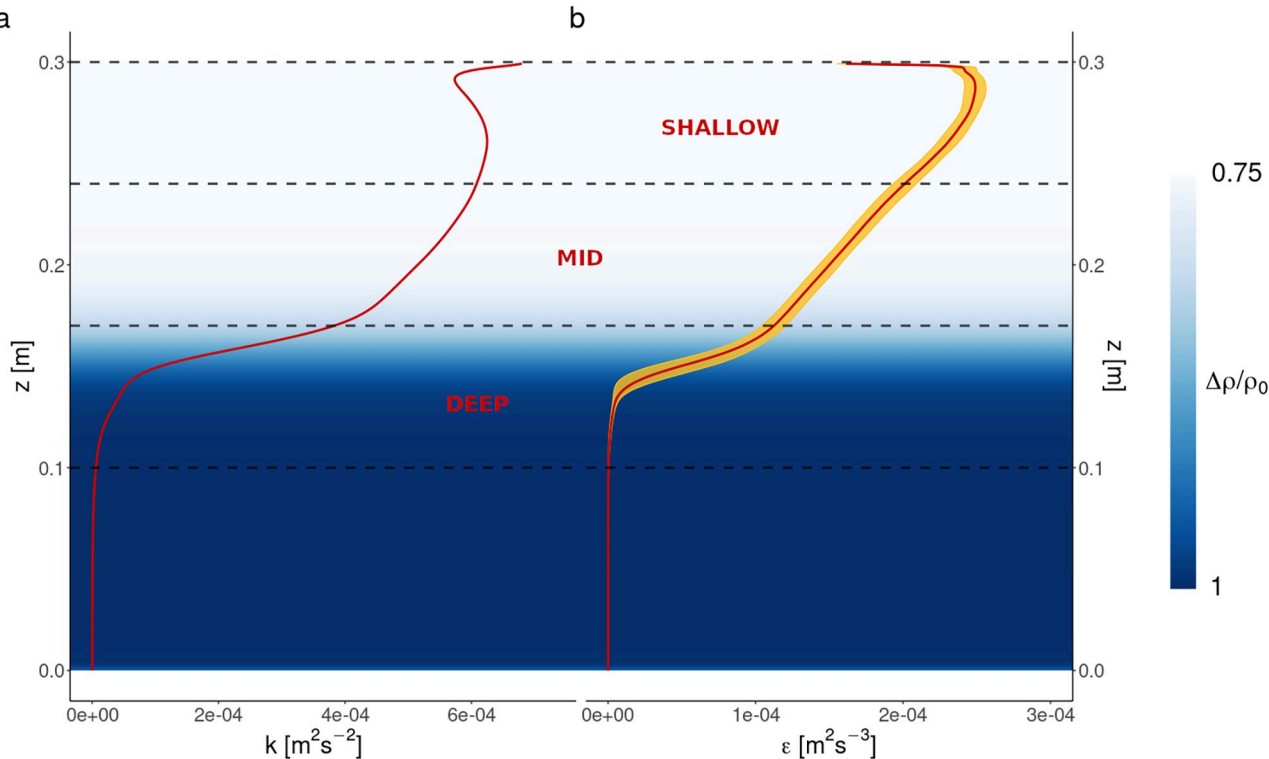

**Fig 1. Depth structure of the simulated flow. The white-blue gradient represents the density gradient of the fluid ($\Delta\rho/\rho_0 = (\rho - \rho_0)/\rho_0$), where $\rho_0$ is the reference density $\rho_0$ at $z = 0$). (a)** Turbulent kinetic energy ($k$) vs depth in the fluid DNS. The red line denotes the mean value of $k$ at all simulated depths. **(b)** Turbulent dissipation rate ($\epsilon$) vs depth in the fluid DNS. The red line denotes the mean value of $\epsilon$ at each simulated depth, while the golden ribbon shows the variance of $\epsilon$ at each depth. Variances are not known for panel (a) since $k$ is itself computed from the variances of the fluid velocities. The Shallow, Mid, and Deep depth regions are labelled in red text and delimited by black dashed lines. Turbulence was strongest near the fluid surface and declined with depth, with essentially quiescent waters below the density interface.

simulation comprised a scaled model of mixed layer turbulence driven by cooling from the surface (see Methods). The spatio-temporal scales of turbulence in this model are relevant, for example, to the effects of cooling during the night, seasonally during the autumn or winter, or during a cold-air outbreak [40–42]. The relative fluid density ($\rho/\rho_0$), the turbulent kinetic energy ($k$) and the rate of turbulent energy dissipation ($\epsilon$) varied with depth and had a weak dependence on time due to deepening of the mixed layer (Fig 1). In particular, turbulence peaked just below the surface of the fluid, and steadily declined with depth before rapidly falling to zero between 0.10 m $< z <$ 0.17 m as it approached the density interface. Below the interface, the fluid was quiescent. We thus expected the dynamics of the microbes to depend on depth. In order to determine the effect of these depth-varying turbulent conditions on microbial patchiness, we divided the parts of the simulation space into which microbes were seeded (see again Methods) into three distinct depth regions: 'Shallow' (0.30 m $\geq z \geq$ 0.24 m), 'Mid' (0.24 m $> z \geq$ 0.17 m) and 'Deep' (0.17 m $> z \geq$ 0.10 m), as shown in Fig 1, and separately analysed patchiness in each depth region. These regions were selected to ignore the deepest ($z \leq$ 0.1 m) part of the simulated flow where microbes were not released and where the fluid was almost entirely quiescent, and to separate the actively turbulent part of the simulated flow into regions of distinct turbulence (very intense turbulence in the Shallow region, a similarly-sized region of declining turbulence in the Mid region, and a sudden drop in turbulence and change in temperature and density in the Deep region, analogous to a thermocline at the bottom of a real-world mixed layer).

### Gyrotaxis and microbe patchiness within the simulated flow

Given the motility parameters $B$ and $v_{swim}$ of a gyrotactic swimmer, and physical parameters of the flow immediately surrounding it like the rate of turbulent energy dissipation $\epsilon$ and the viscosity $v$, earlier studies and simulations have identified two dimensionless numbers that provide insight into the expected degree of patchiness [32, 36]. The stability number $\Psi = B(\epsilon/v)^{1/2}$ is the ratio of of the reorientation timescale $B$ and the Kolmogorov timescale $(v/\epsilon)^{1/2}$, and determines how vulnerable a gyrotactic swimmer is to being overturned by shear. The swimming number $\Phi = v_{swim}/(v\epsilon)^{1/4}$ is the ratio between the microbe's swim speed $v_{swim}$ and the Kolmogorov velocity $(v\epsilon)^{1/4}$ and represents how fast the gyrotactic swimmer can swim relative to the small-scale motion of the surrounding fluid. It is expected that patchiness should be greater among faster swimmers (larger $\Phi$) than among slower swimmers (smaller $\Phi$), and also greater among microbes that can attain a balance between reorientation and turbulent overturning ($\Psi \approx 1$) [32]. Fig 2 shows the range of values of $\Psi$ and $\Phi$ encountered by microbes in our simulations, and thus suggests that faster swimming microbes should exhibit the most patchiness, as should microbes with a reorientation timescale of $B = 1$ s.

We quantified patchiness using the "patch concentration enhancement factor" $Q$ for the 1% most aggregated cells ($f = 0.01$, see Methods) to compare motile and non-motile microbe accumulation into patches. $Q$ is dimensionless, and captures the difference in patch concentration between motile and non-motile microbes. Specifically, $Q$ compares the concentration of motile microbes within patches to the concentration of non-motile microbes within patches. If $Q$ is positive, patches of motile microbes have higher concentrations than patches of non-motile microbes, and vice-versa if $Q$ is negative. The greater the magnitude of $Q$, the greater the difference between the concentration of motile and non-motile microbes within patches.

Comparing the patchiness of the different virtual microbe populations within our simulations, we found that 'agile' motile microbes with fast swim speed (100–500μm s$^{-1}$) and quick reorientation timescale (1–3s) respond differently to turbulence than 'non-agile' microbes of low or intermediate swim speed (10–100μm s$^{-1}$) and slow reorientation timescale (3–5s) (Fig 3). Non-agile motile microbes exhibited little variation in patchiness enhancement at

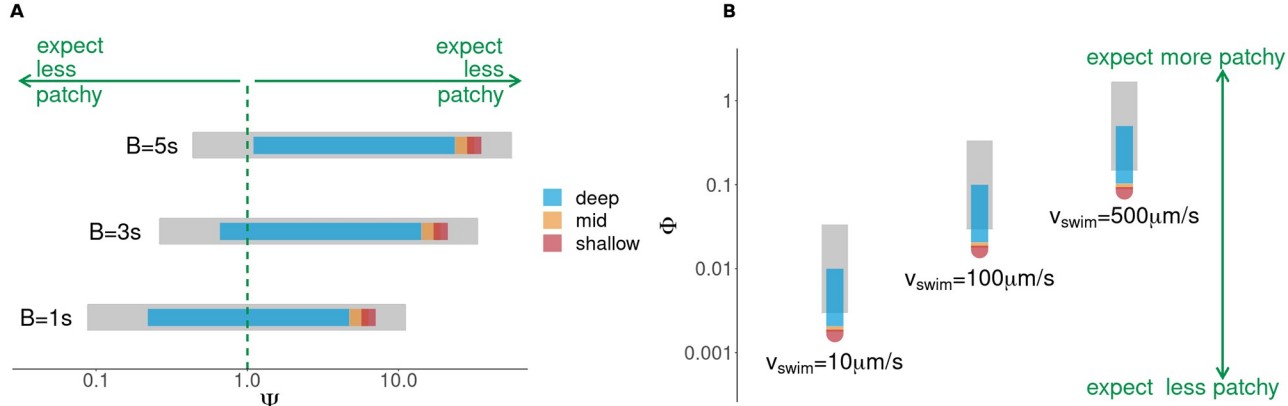

**Fig 2. Dimensionless stability ($\Psi$) and swimming ($\Phi$) number values for microbes at different depths within our IBM. Expected patchiness is greater when $\Psi \approx 1$ and for large $\Phi$. (a)** The stability number $\Psi$ of simulated microbes varied through the Deep (light blue), Mid (yellow) and Shallow (red) depth regions of the flow. To contextualise our simulations with respect to real world flows, grey boxes illustrate the range of the stability number $\Psi$ at expected values of $\epsilon$ and $v$ within a convective oceanic mixed layer (see S2 Text). The stability number $\Psi$ of our microbes is broadly similar to that expected in such real world conditions. **(b)** The swimming number $\Phi$ of simulated microbes also varied through the Deep (light blue), Mid (yellow) and Shallow (red) depth regions of the flow. Here again, grey boxes illustrate the range of the swimming number $\Phi$ at expected values of $\epsilon$ and $v$ within a convective oceanic mixed layer convective turbulence (see again S2 Text). The swimming number of microbes in our IBM overlaps with realistic values, but was generally lower (see Discussion).

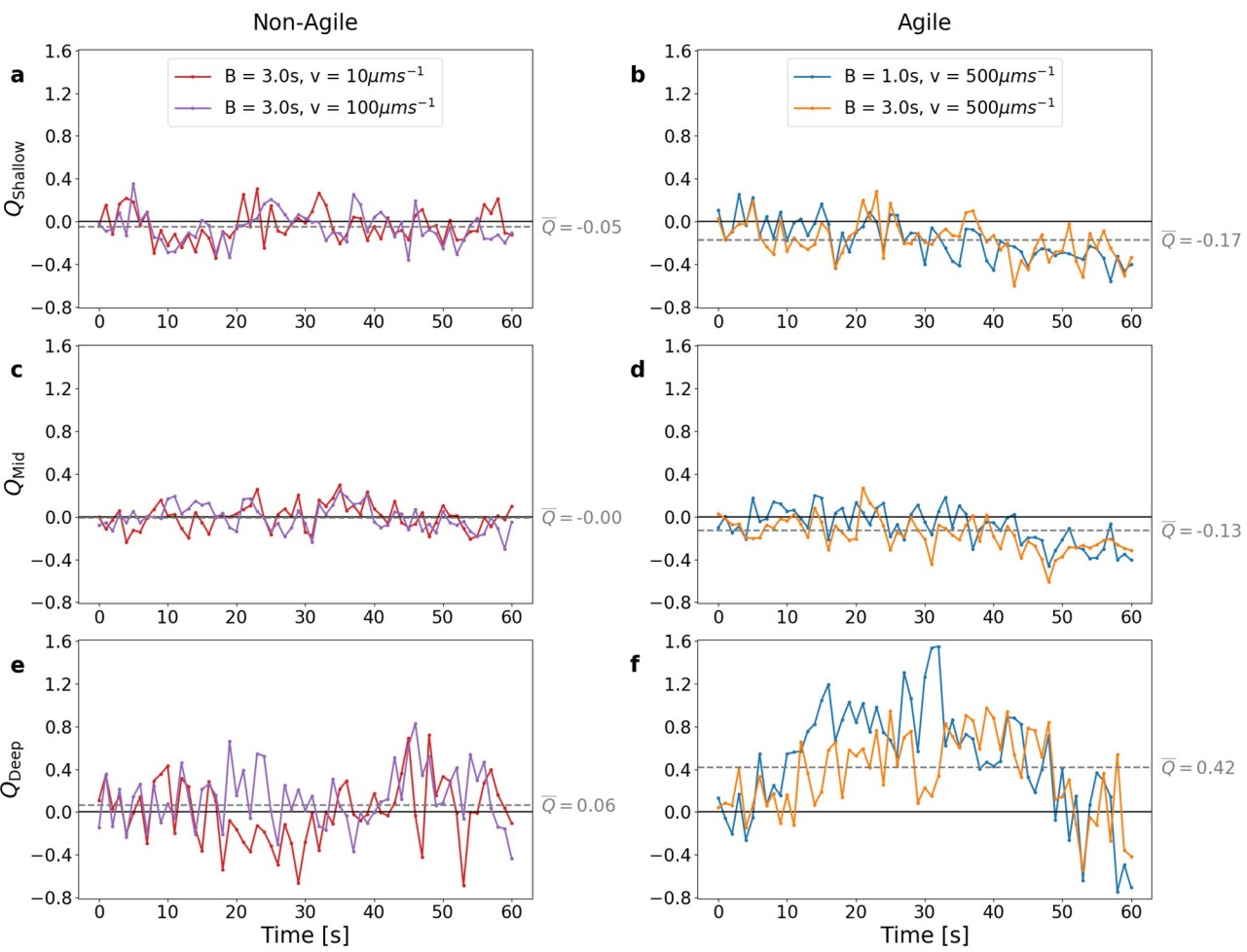

**Fig 3. _Q_-statistic over time (solid lines) in different depth regions.** Subplots in the left-hand column, **(a), (c), (e)**, are from two simulated microbe populations representative of non-agile microbe behaviour. Subplots in the right-hand column, **(b), (d), (f)**, are from two simulated microbe populations representative of agile microbe behaviour. Within each subplot, the dashed gray line represents the mean value $\bar{Q}$ (w.r.t. time) of the $Q$-statistic for the two simulations and depth region plotted therein. Non-agile microbes were not more concentrated in patches than non-motile microbes, whereas agile microbes in the deep region formed patches over twice as concentrated as non-motile microbes ($Q > 1$). Note that agile microbes in the shallower regions exhibited weak but negative mean patch enhancement. Full results for every combination of motility parameters ($B$, $v_{\mathrm{swim}}$) and each depth region are plotted in Figs E, F and G in S3 Text.

different depths (Fig 3a, 3c and 3e). In these populations, patchiness enhancement troughed/ peaked around $Q = -0.4/ + 0.4$ in the Shallow and Mid regions and at about $Q = -0.7/ + 0.8$ in the Deep region, but average enhancement, $\bar{Q}$, was near zero in all three regions, meaning that little mean difference in patchiness was found between these motile microbes and their non-motile counterparts. We note that, although the differences are small, enhancement was generally positive ($\bar{Q}_{\mathrm{deep}} > 0$) in the Deep region of the simulation, but negative ($\bar{Q}_{\mathrm{mid}} < 0$ and $\bar{Q}_{\mathrm{shallow}} < 0$) in the Mid and Shallow regions. As expected, more agile motile microbes exhibited stronger patchiness enhancement (Fig 3b, 3d and 3f). $Q$-values troughed/peaked around $Q = -0.6/ + 0.4$ in the Shallow and Mid regions, and troughed/peaked around $Q = -0.8/ + 1.6$ in the Deep region. Although stronger than in non-agile microbes, average patch enhancement in the Shallow and Mid regions remained weak, and was again strongest in the Deep region.

Overall, two clear trends emerged from this analysis (Fig 4): Firstly, we saw a clear difference between patch enhancement in the Shallow-Mid regions and in the Deep region, with the

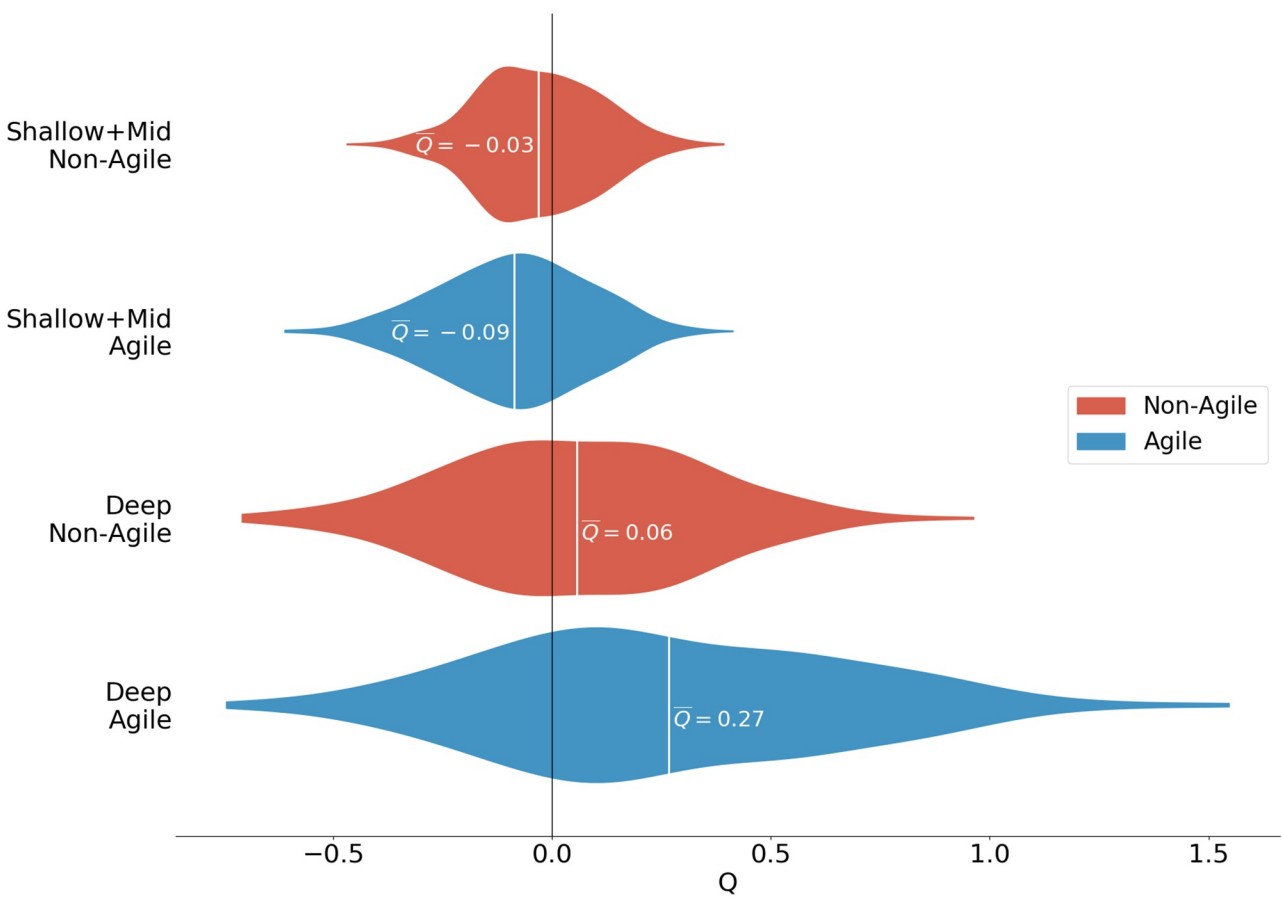

**Fig 4. Violin plot comparison of the distribution of $Q$-values at all times for agile and non-agile microbes in the combined Shallow-Mid regions and the Deep region.** Mean values are marked in white. In the Shallow-Mid regions, motile microbes were not significantly more concentrated in patches than non-motile microbes ($Q \approx 0$). In the Deep region, motile microbes formed patches over twice as concentrated as non-motile microbes ($Q \geq 1$), but non-agile microbes could do so only transiently ($\bar{Q} \approx 0$) whereas agile microbes were consistently more patch concentrated than non-motile microbes ($\bar{Q} = 0.27$).

former generally (and unexpectedly) exhibiting very weak negative patch enhancement ($\bar{Q}_{\text{shallow}} < 0$, $\bar{Q}_{\text{mid}} < 0$) and the latter exhibiting positive patch enhancement ($\bar{Q}_{\text{deep}} > 0$). Fig C in S3 Text shows a top-down view of microbe positions and patches in the Deep region. Secondly, only the most agile motile microbes, sustaining both high swim speeds and fast reorientation timescales, achieved a substantial increase in patchiness from their non-motile cousins ($Q \geq 1$), but then only transiently and in the Deep region. With reductions in swim speed came substantial falls in mean patchiness enhancement ($\bar{Q} \leq 0.07$) (Figs E, F and G in S3 Text).

Virtual microbes with different motility parameters interacted with turbulence to different degrees, stimulating or hindering patchiness in each depth region. Consider the locomotion of individual microbes within the turbulent fluid surrounding them; the microbes experienced a constant battling of forces between the viscous torque exerted on them by shear in the surrounding fluid and the stabilising torque (represented by the reorientation timescale parameter $B$ in our simulations) which reoriented the microbes towards the upwards vertical direction. This was reflected in the orientations of the microbes relative to the vertical in our simulations (Fig 5); microbes with low $B$ reoriented quickly, and were thus more frequently

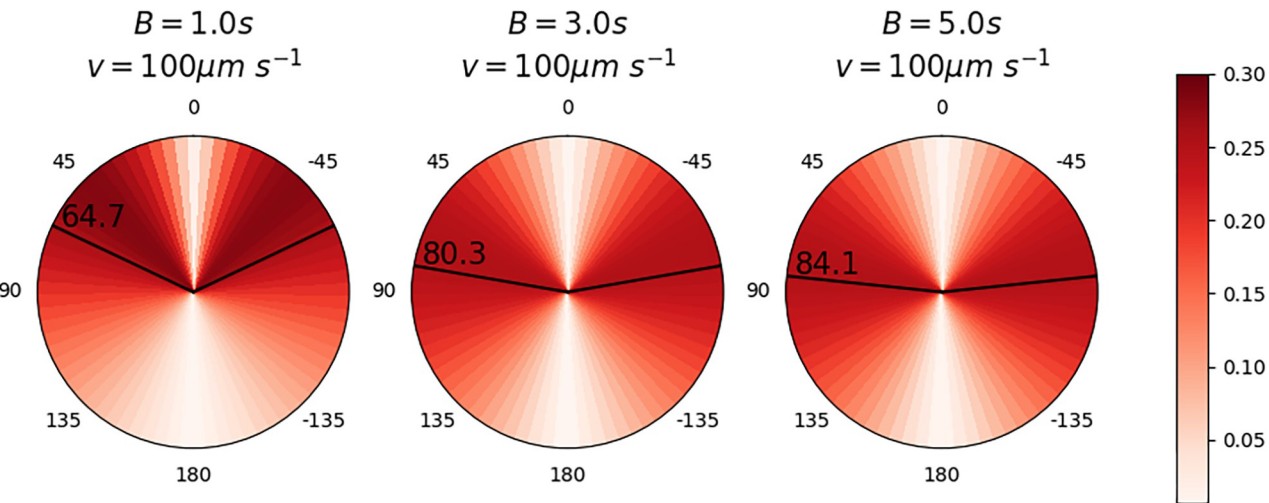

**Fig 5. Normalised distributions of microbe polar angles between 20 − 60s across all depths in three motile simulations with $v_{swim}$ = 100 μm s$^{-1}$ and $B$ = 1s, $B$ = 3s and $B$ = 5s respectively.** A polar angle of 0˚C would represent orientation directly "upwards" towards the fluid surface. Mean polar angles for each simulation are marked and annotated in black. Microbes were subject to a constant balancing between their inherent tendency to orient towards the vertical (captured by the reorientation timescale parameter $B$) and the disorienting effect of turbulence. Faster reorientation (low $B$) resulted in a more vertical orientation than in the case of slower reorientation (high $B$), and slower reorientation also produced more homogeneous orientations. Distributions for all combinations of motility parameters are plotted in Fig H in S3 Text.

able to overcome viscous torque to orient themselves "upwards". Microbes with higher $B$ were slow to reorient and thus more vulnerable to "disorientation" due to viscous torque; their orientations were more homogeneously distributed than those of low-$B$ microbes, and their average orientation was more horizontal. We note that swim speed did not appear to impact the distribution of microbe orientations (Fig H in S3 Text).

Microbe orientation is only part of the story underlying the patchiness trends described above; changes in the spatial distribution of microbes in our simulated flow resulted from the combined effects of individual swimming dynamics and advection by the flow itself. To investigate this combined effect, we computed the 'effective velocity' ($v_{eff}$) of our microbes—the sum of the instantaneous motion of the fluid immediately surrounding a microbe and the microbe's swimming velocity at that moment. We computed the effective velocity in spherical coordinates $v_{eff} = (|v_{eff}|, \theta_{eff}, \phi_{eff})$ since this more naturally yields the magnitude $|v_{eff}|$ ("effective speed") and polar component $\theta_{eff}$ ("effective polar orientation") of the microbes. In particular, the polar component $\theta_{eff}$ captures the tendency of microbes to alter their depth within the simulated fluid, unlike the microbe orientation, which does not account for the movement of the surrounding fluid. Similarly, the magnitude of the effective velocity captures the speed at which microbes moved through the simulated space, not just their motion relative to that of the surrounding fluid. Within each simulation we computed effective velocities for all microbes every 1 second from $t$ = 20–60s and plotted the distributions of their magnitude and polar orientations (Fig 6). As with the preceding analyses, we restrict our focus to examining $v_{eff}$ separately in the Shallow, Mid and Deep regions.

Within the Shallow (Fig 6a and 6b) and Mid (Fig 6c and 6d) regions, effective velocities exhibited little variation between simulations, with both the polar angle and the magnitude of effective velocity remaining relatively constant across the full range of $B$ and $v_{swim}$ parameter values. In the Deep region (Fig 6e and 6f), the polar angle was approximately horizontal but again exhibited little variation between simulations, while the magnitude varied by a margin ($\sim 13$ mms$^{-1}$) considerably greater than the largest difference in microbe swim speeds

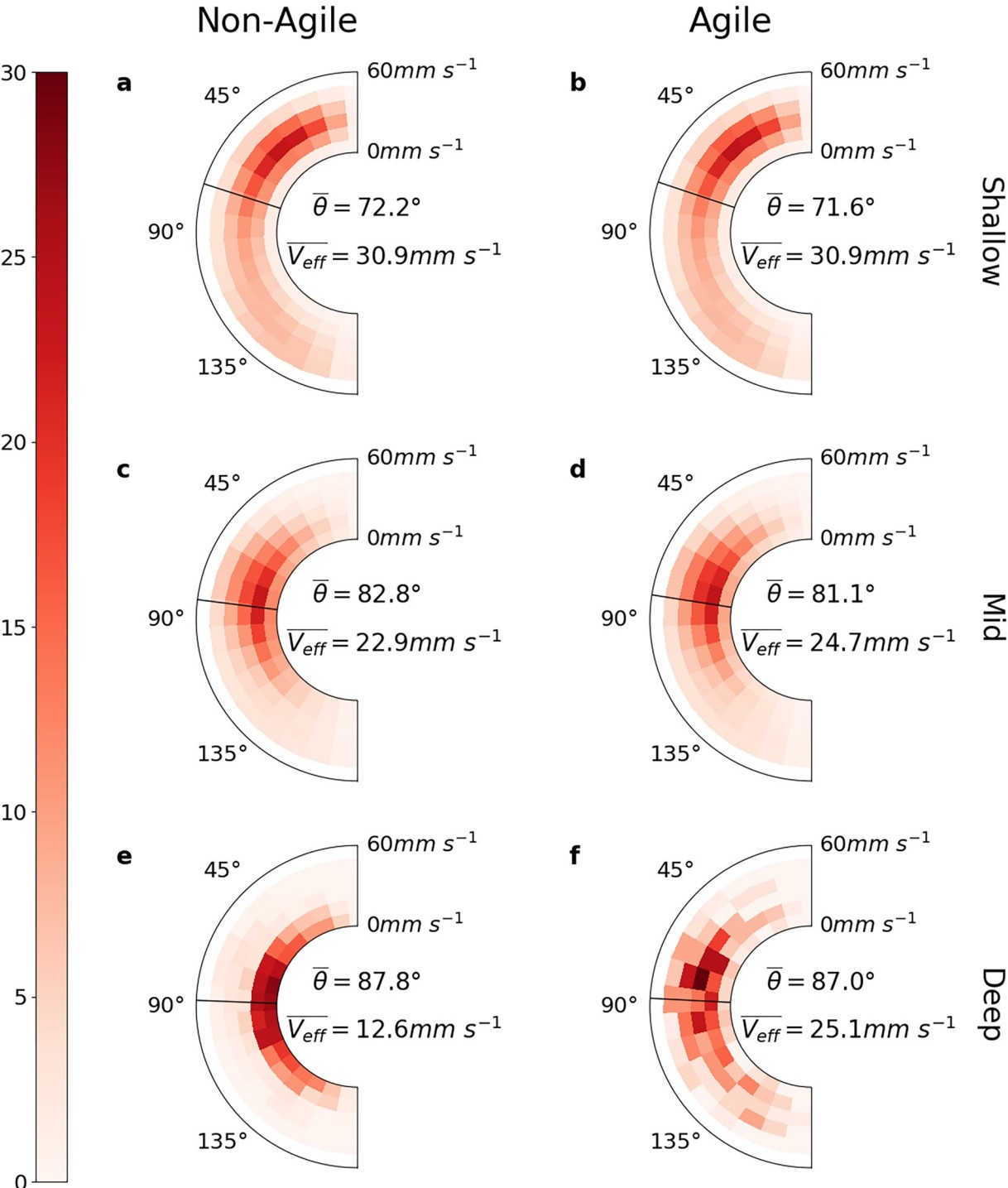

**Fig 6. Normalised distributions of the magnitude and polar angle of effective velocity in each depth region of two simulations, respectively characteristic of non-agile (a, c, e) and agile (b, d, f) microbes.** Microbes in the deep region had near-horizontal effective velocity, which acted to restrict their movement in the vertical direction. Also in the deep region, the difference in the magnitude of effective velocity ("effective speed") was many times larger than the difference in microbial swimming speed. Elsewhere, effective speeds were very similar between all simulations, and the effective velocity was less vertically constrained.

between simulations (0.49 mms$^{-1}$). Overall, at greater depth, effective velocity slowed and became increasingly horizontal, with microbes in the 'Deep' region moving near-horizontally.

## Discussion

Microscale microbe patchiness may have far-reaching implications both for the microbes themselves and for the wider ecosystem, but it is first and foremost essential to accurately understand the prevalence and intensity of such patchiness in realistic conditions. By combining state-of-the-art individual-based modelling tools with a scaled-down 3D model of convective turbulence, our study investigated whether and how turbulent fluid motion can trigger increased patchiness through a coupling of vortical fluid shear and microbial motility. Given the motility parameters $B$ and $v_{swim}$ of a gyrotactic swimmer, the dimensionless stability number ($\Psi$) and swimming number ($\Phi$) provide insight into the expected outcome of this coupling. Previous studies in more idealised models of turbulence [32, 36] concluded that patchiness should be greatest among fast swimmers (large $\Phi$) and among microbes that can attain a balance between reorientation and turbulent overturning ($\Psi \approx 1$). Although this predicts greater patchiness at larger swimming numbers, we took care to limit our simulations to microbes with realistic swimming speeds ($v_{swim} \leq 500$ μms$^{-1}$) to keep the study biologically relevant. The simulations in this study cover a substantial range of values for the gyrotaxis parameters $B$ and $v_{swim}$ from the literature, rather than simulating specific individual species with known and measured swim speeds. The maximal simulated 500 μm s$^{-1}$ swim speed should thus be interpreted as an upper limit for microbes, which we used to explore as fully as possible how swimming might produce patchiness via turbulent interactions. We found that, in a mixed layer undergoing convective overturning, patchiness was limited to highly motile microbes and to deeper regions of the mixed layer, where turbulence is less intense than near the surface. This is consistent with the aforementioned studies whose predictions we validate for this more complex flow regime. Note that the $Q$ statistic captures "excess" patchiness among motile microbes relative to non-swimmers. To clarify the overall importance of motility to the emergence of patchiness we also computed "absolute" concentrations of microbes in the Deep regions of all of our simulations during the 10-second windows (determined by inspection) when $Q$ was maximal or minimal for each simulation (Figs J, K, L M, N, O, P, Q and R in S3 Text). These plots clarify that the distributions of absolute microbe concentration in patches for the non-motile simulation are generally distinct from that of the motile simulations, and that when $Q$ is maximal the distribution for the non-motile simulation has a lower central tendency than that of the motile simulations (and vice-versa when $Q$ is minimal).

Although our DNS simulates only a relatively small physical volume of water, physical scaling arguments (see S2 Text) justify the applicability of the DNS to our investigation of turbulence-driven patch enhancement. The key quantity is the Kolmogorov timescale ($\tau_K$), which in our simulation is between 0.137–9.89s (varying with depth as the turbulent energy dissipation varies). For comparison with real world flows, this sits comfortably within the $\sim$ 0.1–10s range expected of ocean mixed layer conditions (see again S2 Text for details). Scaling arguments also establish limits on the differences in the intensity of turbulence between our DNS and larger real-world flows, demonstrating that turbulent velocity fluctuations in a convective ocean mixed layer are between $0.93$ to $7.88$-fold stronger than in our DNS (S2 Text). To interpret our findings in a real world context, we employ these scaling relationships together with the non-dimensional swimming and stability numbers, in the discussion below. Future efforts to model microscale spatial activity in realistic flow regimes, including turbulence, would benefit from detailed empirical measurements of how turbulent fluid velocity fluctuations vary from small-scale volumes such as our DNS, to real-world scenarios such as the marine water

column. This could substantially simplify the process of determining how to interpret the results and predictions of scaled-down models to real-world systems.

Convective overturning is a key driver of mixed-layer turbulence [43–45]. Our buoyancy-driven DNS reproduces convective mixing driven by heat loss from the fluid surface. Comparable oceanic conditions are most commonly associated with the mid-latitudes, though particularly strong and deep convection can also occur in sub-polar regions such as the Labrador and Greenland seas [46]. Surface cooling varies both daily, with the solar cycle, and seasonally, with oceans in particular acting as heat reservoirs during autumn and winter [42]. Our simulations are therefore best interpreted in the context of a body of water undergoing convective mixing due to heat loss to the atmosphere during autumnal or winter cooling, during the night, or during a cold-air outbreak [40, 41]. It must be stressed that the ocean mixed layer is not constantly overturning, but also undergoes periods of minimal or negative heat loss through the surface, when our simulated turbulence regime is not applicable.

We analysed patchiness separately in the Shallow, Mid and Deep regions of our simulation. This is important since our DNS does not reproduce all of the dynamics of a real world mixed layer. In particular, because our simulation's mixed layer depth is small ($\sim 0.15$m), the time-scale at which convective motions travel the full distance from the surface to the bottom of our mixed layer is short by comparison to, for example, an oceanic mixed layer. We therefore focus on how local turbulent conditions drive patchiness within the 3 depth regions that we have addressed rather than deriving results about the large-scale vertical movement of microbes through the full simulated water column. Our results suggest that in a convective mixed-layer, turbulent fluid motion near the surface of the fluid will greatly exceed the locomotive capabilities of motile microbes, inhibiting patch enhancement. At depths farther from the fluid surface, turbulence is relatively weak, analogously to the region near or below the thermocline in a real body of water. At these relatively quiescent depths, highly agile microbes can attain the balance of viscous and stabilising torques ($\Psi \approx 1$) that drives patch enhancement, forming patches over twice as concentrated as non-motile microbes ($\max(Q) \approx 1.6$). In oceanic flows with particularly strong surface cooling (and hence a deeper mixed-layer), turbulent velocity fluctuations can be up to 8-fold stronger than in our simulation (see again S2 Text), and patch enhancement may be more difficult to achieve than in our simulations. On the other hand, comparing the swimming numbers $\Phi$ experienced by microbes in our simulations, to those expected in a comparable oceanic mixed layer, suggests that patchiness in a real mixed layer may be stronger than in our simulations (Fig 2B). Due to computational constraints, we have not modelled the effects of other sources of turbulence, such as wind or waves. We predict that, since turbulence from these sources would also be strongest near the fluid surface and decline with depth, our results would not qualitatively change with their inclusion; additional turbulence near the fluid surface would continue to disperse microbe patches, while at greater depth, highly motile microbes may begin to form concentrated patches through coupling with weaker turbulence.

Since the patchiness metric $Q$ is fundamentally a measure of the difference in patchiness between motile and non-motile cells (see Methods), caution must be applied when comparing values of $Q$ in our simulations to those of previous studies of patchiness within turbulent flows in a statistically steady state, wherein non-motile cells may aggregate differently than in our simulations. For example, prior studies employing idealised steady-state isotropic turbulence models have assumed that $Q = 0$ is a lower bound for patchiness [32]. Negative Q values in our simulations are therefore surprising, as they suggest that the most patch-concentrated motile microbes can be less clustered than the most patch-concentrated non-motile microbes. One explanation for this behaviour in our simulations is that, within the patches formed by the fraction $f$-most aggregated microbes, motility may occasionally be oriented away from the center

of the patch, during which time active swimming would reduce clustering relative to corresponding patches of non-motile microbes.

Our analysis of the "effective velocities" of our simulated microbes is consistent with the hypothesis [32] that positive patch enhancement is caused by the interaction of turbulent fluid vorticity and the stabilising torque of the microbes themselves. In regions of positive patch enhancement, the net effect of these competing forces is to constrain average microbe movement to a nearly horizontal direction (Fig 6e and 6f). The Deep region of our simulated flow is where this effect is strongest. Although gyrotactic swimmers in this region are constantly attempting to orient their swimming direction towards the vertical, their effective direction of motion is still dominated overall by local fluid motion, since this is stronger in the horizontal direction than their swimming is in the vertical direction (Fig Y in S3 Text). This may be the proximate cause of positive mean patch enhancement; with the vertical dimension effectively denied to them, microbes move in an horizontal sub-space, increasing local concentrations within that subspace relative to unconstrained non-motile microbes. To explicitly test the importance of the coupling of gyrotactic motility with turbulence, as opposed to generic motility within turbulence, we performed two additional simulations involving slow-swimming and fast-swimming non-gyrotactic motile microbes (i.e. microbes without a gyrotactic reorientation parameter $B$, whose orientations were therefore exclusively determined by the effect of turbulence). In contrast to their gyrotactic motile counterparts, these non-gyrotactic but still motile microbes did not exhibit significant patch enhancement in any region of the flow (Fig S in S3 Text). Analysis of their effective velocities (Fig T in S3 Text) suggests that in the absence of gyrotactic reorientation to counterbalance the effect of turbulent fluid vorticity, average non-gyrotactic motile microbe movement is not as constrained to the horizontal direction as it is for gyrotactic motile swimmers. These additional simulations demonstrate the importance of a reorientation mechanism for triggering enhanced patchiness within turbulence, in particular an anisotropic reorientation such as gyrotaxis. However many microbial species, particularly bacteria, exhibit isotropic reorientation mechanisms (e.g. "tumbles", "flicks" or Brownian rotational motion), which may interact quite differently with turbulence than the gyrotactic motility modelled here. Although beyond the scope of this study, these alternative mechanisms present a clear opportunity for future simulations investigating interactions between small-scale turbulence and microbial motility.

Finally, our results raise interesting questions about the utility of gyrotactic locomotion in different turbulent conditions. In the presence of strong turbulence in particular, increased agility has little effect on the effective velocity of a microbe (Fig 6a, 6b, 6c and 6d), and only a very weak (and negative) effect on patch enhancement (Fig 4). In more quiescent waters, we saw up to a 2-fold ratio between the effective speeds ($|v_{\text{eff}}|$) of agile and non-agile swimmers. Fig I in S3 Text suggests that the mechanism for this large increase in effective speed is that agile swimmers are more efficient than their less agile counterparts at encountering and remaining within fast-moving packets of fluid, allowing these fast microbes to boost their effective velocity by up to 12 mm s$^{-1}$ in our simulations—an increase well beyond the physiological capabilities of microbial motility alone [47, 48]. Interestingly, in our additional simulations of non-gyrotactic motile microbes (Fig T in S3 Text) we did not observe a similar velocity-boosting effect among fast swimmers, suggesting that coupling of gyrotaxis with turbulence is critical to remaining within fast-moving packets of fluid. Furthermore, motile microbes are known to modify their swim speed in response to environmental stimuli such as temperature [49], nutrient concentration gradients [50, 51], and even turbulence [52, 53], and thus could strategically vary their agility in response to local fluid and environmental conditions. For example, microbes in patch-enhancing turbulence could downregulate swim speed to decrease patchiness and reduce predation [21] or viral infection risks [22, 23], and microbes

in shallower, nutrient-poor waters could temporarily upregulate swimming speed in order to improve their chances of entering a downwell towards deeper waters where nutrients are typically more plentiful [54].

Although our fully resolved 3D turbulent DNS permits the investigation of turbulence-motility coupling and its effect on microbial patchiness in substantial detail, the spatial and temporal resolution of the simulation that allows this also engenders limitations on the length of the simulations that can feasibly be carried out. Our simulations generated microbial trajectories spanning 60 s from an initially random distribution (in space and in the orientation of the motile microbes). This was sufficient for the microbes to produce patches and to detect differences in patchiness among the different populations of microbes we simulated. To determine the robustness of our results to the total simulation time, and demonstrate that the results of this study are not merely a consequence of the simulation length, we repeated our $Q$ statistic analysis on truncated subsets of our simulated data. Specifically, we truncated the simulations to end after $t$ = 20 s, 30 s, 40 s, and 50 s, and calculated the distributions of $Q$ values found for each truncated simulation (Figs U, V, W and X in S3 Text). We found that, despite the greater tendency for negative $Q$ later in the simulations, the qualitative nature of our results does not vary between the truncated and the full-size simulations; in all cases patch enhancement in the Shallow-Mid regions was generally weak and negative, and stronger and positive in the Deep region, particularly for "agile" microbes—these conclusions appear not to depend on the simulation time, at least within the range of times 0–60s. Nonetheless it remains possible that over longer timescales, the dynamics of patches and of patch formation may differ from what we observed in this study. For example it can be seen from Fig 3 (and Figs E, F and G in S3 Text) that some of our simulated populations exhibit first an upwards trend in $Q$, followed by a downwards trend in $Q$ towards the end of the simulations. Although in turbulent conditions, individual patches are likely to persist only for relatively short timescales, the long-term behaviour of $Q$, which captures differences in overall patchiness between non-motile and gyrotactic microbial populations, remains a salient and open question. $Q$ might behave periodically, oscillating between positive and negative values with a period $\geq$ 60 s, or alternatively might settle to a steady zero or non-zero value over time, in which case the results presented in this study may prove to apply mostly to patch creation, rather than patch maintenance over time. Clarifying this open question by determining the long-term behaviour of $Q$ in realistic turbulence regimes may be a fruitful target of future research, since extending our simulations temporally is beyond the reach of the present study without surrendering the crucial temporal and spatial resolution that underpins it. Additional targets of future research on the question of turbulence-driven patchiness include investigating the relationship between the tendency of microbes to form patches and their ability to navigate the water column more broadly by exploiting fast-moving packets of fluid, as we report in this study. The nature of such "expressways" is of interest in its own right (for example their directionality, their longevity, and the precise mechanism(s) by which agile microbes encounter them more efficiently than non-agile microbes) that may now warrant its own investigation.

Our results demonstrate that turbulence-driven microscale patchiness is a delicate balancing act of physical fluid conditions and individual motility, and not a ubiquitous consequence of gyrotaxis. There is much scope for further individual-based modelling at these scales to further expand our understanding of the interactions at play; for example by dynamically tuning motility in response to environmental cues, or by incorporating additional trophic levels such as nutrients or predators. The longevity of patches, when they do occur, is a key additional component of their ecological importance; in order for microbe patches to affect reproduction or nutrient distributions, for example, the lifetime of a patch cannot be smaller than the time-scale of reproduction or of resource consumption [9]. Since we do not currently have the tools

to reliably determine patch lifetimes for comparison between simulations, we are not able to report patch lifetimes for the results from this study. Modelling or measuring in-situ the longevity of microbe patches, and their longevity's dependence on conditions such as turbulence, will be critical to future research on this topic.

## Materials and methods

Our microbe simulations consisted of a series of experiments in a fully resolved lab-scale simulation of convective mixing due to surface cooling in a two-layer stratified flow. Each experiment targeted a different combination of biological parameters controlling the motility of the microbes, in order to test the predictions of earlier work and understand how sensitive patchiness is to motility parameters. In this section we give details of the fluid simulation, the microbe IBM and the algorithms used to quantify patches and patchiness.

### Direct simulation of a mixed layer driven by convection

The flow targeted in this paper is convection in the top-layer of a two-layer stratified fluid due to surface cooling, of the kind that produces convective mixing in the oceans and lakes [43–45]. As we aimed to simulate the behaviour of microbes, it was essential that all dynamic scales of the turbulence be resolved. This implies (1) that a highly accurate code for direct numerical simulation needed to be employed; and (2) that the problem should be scaled down to a lab-scale, since it is impossible to resolve all the turbulence at real-world (e.g. oceanic) scales.

The domain was $0.6 \times 0.6 \times 0.3$ m (length $\times$ width $\times$ height), and the fluid inside the domain comprised two layers of thickness $h_0 = 0.15$ m of which the top layer had a density 75% lower than that of the bottom layer, thus creating a stable stratification. The density jump used here is much stronger than in lakes or the ocean, but was chosen to limit turbulent entrainment and thus slow down the deepening of the mixed layer [55]. The strength of the density jump does not significantly affect the turbulent flow in the mixed layer, except very close to the interface. The fluid was subject to a stochastic initial condition and a negative buoyancy flux $\mathcal{B}$ at the top of the domain, which is representative of the cooling of the water surface due to long wave radiation during night time, winter or autumn cooling, or during a cold-air outbreak. As a result, the fluid near the surface cooled and began to descend, forming a convective mixed layer above the density interface (Fig 7).

In order to be able to resolve all the turbulent scales of motion, a kinematic viscosity $v = 5 \times 10^{-6}$ m$^2$ s$^{-1}$ was chosen, which is slightly higher than that of water. A thermal diffusivity was set to $4 \times 10^{-6}$ m$^2$ s$^{-1}$. Setting the buoyancy flux $\mathcal{B} = 5 \times 10^{-4}$ m$^2$ s$^{-3}$ implies a characteristic velocity scale of the mixed layer $w_* = (\mathcal{B}h_0)^{1/3} = 0.042$m s$^{-1}$ [56, 57] (see S2 Text). This implies that the initial bulk Richardson number and Reynolds number are:

$$Ri_0 = \frac{h_0 \Delta b}{w^{*2}} = 85 \;, \qquad Re = \frac{w^* h_0}{v} = 1260. \qquad (1)$$

The computational grid was $720 \times 720 \times 360$, which corresponds to 186,624,000 cubic cells. Cell side-lengths were thus $\Delta x = \Delta y = \Delta_z \approx 0.83$ mm. The dissipation rate peaked at $\epsilon = 2.66 \times 10^{-4}$ m$^2$ s$^{-3}$ (see Fig 1), which implies that the Kolmogorov length scale is $\eta_K = (v^3/\epsilon)^{1/4} = 0.828$ mm. This is the size of the smallest turbulent eddy that is encountered in the flow. Since $\Delta x/\eta_K \approx 1$, it follows that all the turbulent scales of the flow were resolved, and the simulation can indeed be considered DNS.

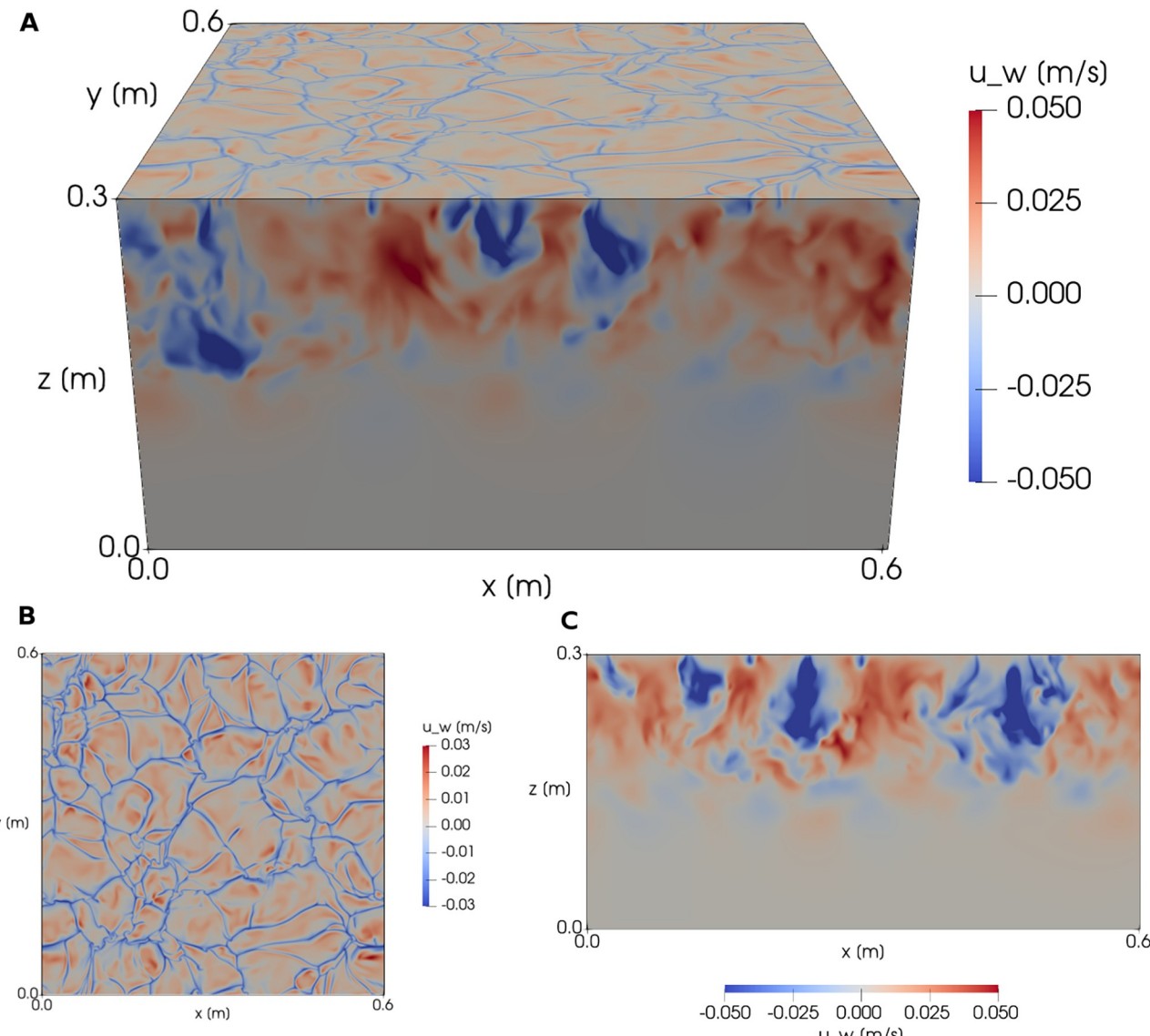

**Fig 7. Vertical fluid velocities (in ms⁻¹) at *t* = 60 s during the SPARKLE DNS.** Axes are labelled in units of DNS cell side-length. **(a)** shows the velocities at the surface and sides of the simulation. **(b)** is a top-down view of the fluid surface demonstrating rising regions of less dense fluid (red) pushing aside denser, falling fluid (blue). **(c)** is a side-on cross-section view through the center of the simulation, demonstrating the depth profile of these cooler (blue) and warmer (red) regions.

At *t* = 0s, the fluid was quiescent, after which the buoyancy flux was switched on. The total simulation time was 90 s, of which the first 30 s are 'spin-up' time, in which the convective mixed layer was formed. The microbe simulations commenced after these initial transients, and thus only used the data for $30 \leq t \leq 90$ s.

The velocity and density fields were obtained using the direct numerical simulation code SPARKLE, which employs a symmetry-preserving fourth-order-accurate finite volume discretization scheme, preserving mass, momentum and energy [58, 59]. No modifications to SPARKLE were implemented for the purposes of this article. SPARKLE solves the Navier-Stokes

equations in 3D in the Boussinesq approximation:

$$\nabla \cdot \boldsymbol{u} = 0,$$

$$\frac{\partial \boldsymbol{u}}{\partial t} + \nabla \cdot (\boldsymbol{u} \otimes \boldsymbol{u}) = -\nabla p + \nu \nabla^2 \boldsymbol{u} + b\boldsymbol{k},$$

$$\frac{\partial b}{\partial t} + \nabla \cdot (\boldsymbol{u}b) = \kappa \nabla^2 b,$$

(2)

where space is denoted as $\boldsymbol{x} = (x, y, z)$ and fluid velocity by $\boldsymbol{u} = \boldsymbol{u}(\boldsymbol{x}) = (u(\boldsymbol{x}), v(\boldsymbol{x}), w(\boldsymbol{x}))$, $p$ is kinematic pressure, $\nu$ is kinematic viscosity, $b = b(\boldsymbol{x}, t)$ is buoyancy, $\kappa$ is thermal diffusivity and $\boldsymbol{k}$ is the unit vector in the $z$-direction. The buoyancy $b(\boldsymbol{x}, t)$ is given by a linear equation of state $b = \beta g\theta$, where $\beta = -T_0^{-1}\partial\rho/\partial T|_{T_0}$ is the expansion coefficient, $g$ is the gravitational acceleration and $\theta = T - T_0$ is the temperature relative to the reference temperature $T_0$.

## Gyrotactic microbe IBM

Since aquatic environments are in constant motion, microbial ecosystems involve many inherently Lagrangian processes that are well-suited to individual-based models (IBMs). IBMs are particularly appropriate for explicitly modelling 3D aquatic ecosystems in complex flow regimes, wherein agents must interact individually with their local environment (a turbulent eddy, for instance, or a nutrient patch), and/or with each other, and where complex ecosystem dynamics can emerge naturally from the collective behaviour of individuals in the model. IBMs of this kind have already seen active service in ecological research pertaining to questions as diverse as microbial patchiness [32] and evolutionary dynamics [60], spatial dynamics of fish [61], fish larvae [62] and sea turtle hatchlings [63], thermal responses in phytoplankton populations [64] and the dynamics of ocean plastics [65–67]. Here we describe the mathematical framework of our microbial motility model and its implementation using the OceanParcels [68, 69] Lagrangian analysis toolkit.

We adopted the "gyrotaxis" model of motility, a common microbial strategy for controlling vertical motion by swimming with a stabilising torque that acts to continually bias the swimming direction towards the surface [70]. Consider the microbes as spheres with off-set centres of gravity, such that they passively align with the vertical in the absence of any external forces. Microbial movement was then modelled as two concurrent processes—advection by the turbulent flow (for all simulated microbes), and individual locomotion in the instantaneous swimming direction (for motile microbes only). The re-orientation of microbes under turbulence and their subsequent re-alignment with the vertical is governed by the following equation [70]:

$$\frac{d\boldsymbol{p}}{dt} = \frac{1}{2B} \left[\boldsymbol{k} - (\boldsymbol{k} \cdot \boldsymbol{p})\boldsymbol{p}\right] + \frac{1}{2}(\boldsymbol{\omega} \times \boldsymbol{p}),$$

(3)

where $\boldsymbol{p}$ is a unit vector describing the swimming direction, $\boldsymbol{\omega} = \nabla \times \boldsymbol{u}$ is the fluid vorticity (curl of the velocity field), $\boldsymbol{k} = (0, 0, 1)$ is the unit vector in the positive vertical direction, and $B$ is the 'gyrotactic reorientation timescale' describing the typical time required for a disoriented cell to return to vertical alignment if $\boldsymbol{\omega} = 0$. Under this framework, the first term on the RHS encodes a microbe's reorientation towards the vertical, while the second term encodes the 'overturning' effect of turbulence on the microbe, due to viscous torque. Since the 'orientation' of the microbes is determined by their swimming direction $\boldsymbol{p}$, non-motile microbes in our simulations had no orientation. Motile microbe swimming velocities and gyrotactic reorientation timescales ($B$) were set as constant within each simulation, in order to test the sensitivity of patch formation to these biological parameters.

**OceanParcels—Computing microbe trajectories.**   We computed motile and non-motile particle trajectories using the OceanParcels Lagrangian particle tracking engine. Velocity fields were pre-computed in SPARKLE and fed to OceanParcels (version 2.1.4) as netCDF4 files. We did not consider the physical effects of microbes on the flow. No modifications to the Ocean-Parcels source code were made for the purposes of this article. We used OceanParcel's "custom kernel" functionality to track microbes by integrating the per-microbe velocity associated with the superposition of the microbial swimming and the flow at each timestep:

$$\frac{d\boldsymbol{X}}{dt} = \boldsymbol{p}v_{\text{swim}} + \boldsymbol{u}(\boldsymbol{X}), \tag{4}$$

where $\boldsymbol{X}$ is the microbe's position, $v_{\text{swim}}$ is the swimming velocity of the microbe and once again $\boldsymbol{p}$ is its swimming direction and $\boldsymbol{u}$ is the fluid velocity. Microbes in the non-motile simulation had their $v_{\text{swim}}$ set to 0 µm s$^{-1}$, so that their movement was exclusively determined by the motion of the surrounding fluid. This approach has previously been shown to accurately capture the trajectories of passive and active swimmers in a turbulent flow [32]. The microbe positions $\boldsymbol{X}$ were integrated with an RK4 method, and the swimming directions $\boldsymbol{p}$ were integrated with a Forward Euler method. All simulation outputs were stored in the netCDF4 file format.

In each simulation, 100,000 microbes were initialised at random positions within the upper half ($z \geq 0.15$ m) of the DNS flow (above the density interface), and with a random swimming direction. For clarity, no other randomness or stochasticity was introduced into the microbial IBM. Microbes were not initialised below the interface since the flow there is quiescent, lacking sufficient turbulent motion to drive patch enhancement. Periodic boundary conditions were applied in the horizontal directions, as in the DNS, and a reflective boundary condition was applied to the top (surface) boundary. Fig 8 shows a snapshot of microbe positions and a subset of microbe trajectories during one of the simulations. Each simulation ran from the end of the DNS spin-up period to the end of the DNS (60 s in total), with a timestep of $\Delta t = 0.01$ s. The choice of timestep is discussed in detail in S1 Text. We recorded the position and swimming direction of each microbe every 0.1$s$. Values for the parameters $B$ and $v_{\text{swim}}$ were chosen to span the range of values for these parameters estimated in the existing literature [71–73],

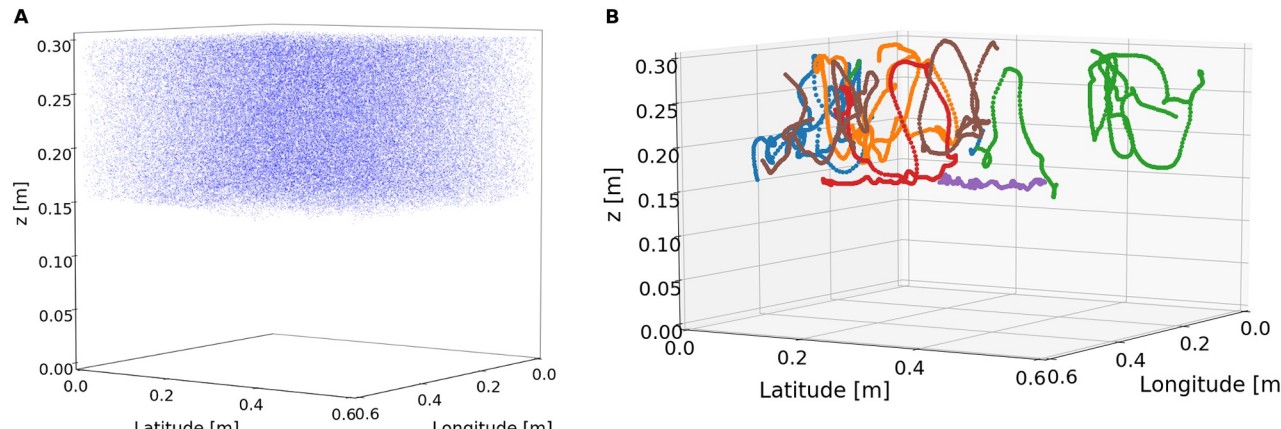

**Fig 8. Microbial motion within the simulated flow. (a)** A snapshot of microbe positions at $t = 25$ s of the $(B, v_{\text{swim}}) = (5$ s, 10 µm s$^{-1})$ motile simulation. **(b)** Sample of six 3D microbe trajectories from t = 0–60s in the $(B, v_{\text{swim}}) = (5$ s, 10 µm s$^{-1})$ motile simulation. Each uniquely-coloured sequence of dots represents a single microbe's trajectory. Owing to the periodic boundaries in the longitudinal and latitudinal directions, trajectories may appear discontinuous when a microbe moves through such a boundary (e.g. green trajectory). Microbes spent time in each of the three depth regions considered in our analysis, mostly due to advection by the surrounding fluid, but also through individual locomotion in less fast-moving regions of the fluid. In particular, long sojourns were noticeable at greater depths where turbulent fluid motion is less intense.

**Table 1. OceanParcels simulation parameters.**

| Parameter | Units | Value |
|-----------|-------|-------|
| Dimensions | m | $0.6 \times 0.6 \times 0.3$ |
| Total time | s | 60 |
| $\Delta t$ | s | 0.01 |
| $n_{particles}$ | – | 100,000 |
| $B$ (motile) | s | 1.0, 3.0 or 5.0 |
| $v_{swim}$ (motile) | μm s$^{-1}$ | 10, 100 or 500 |
| $B$ (non-motile) | s | – |
| $v_{swim}$ (non-motile) | μm s$^{-1}$ | 0 |

rather than simulating specific individual species with known and measured swim speeds. This comprises a rather substantial range of values, particularly for the microbial swim speed $v_{swim}$, whose maximal value of 500 μm s$^{-1}$ in the simulations should be interpreted as a upper limit for microbial swimming. Table 1 summarises the parameters for the OceanParcels simulations.

## Quantifying patchiness

**Voronoi tessellation.**   In order to obtain a diagnostic for the local microbe concentration, we performed a 3D Voronoi tessellation of the microbe positions every second using the Voro++ package (version 0.4.5), applying the same double horizontally periodic boundary conditions as the simulations and specifying upper ($z = 0.3$ m) and lower ($z = 0$ m) boundaries as in the simulated space. The Voronoi tessellation assigns to each microbe the polyhedral containing all points in the simulated space that are closer to that microbe than any other. The inverse of the volume of this polyhedral gives a measure of the local microbe concentration within that polyhedral.

**Q-statistic.**   To quantify patchiness in our simulations, we first performed the 3D Voronoi tessellation described above. Then, following the approach outlined in [32], we defined patches to consist of the fraction $f$ of microbes with the largest local concentration (i.e. smallest Voronoi polyhedron volume), and we used the concentrations within these patches to calculate, at every second, the 'patch concentration enhancement factor' $Q$:

$$Q = \frac{C - C_P}{C_M}, \tag{5}$$

where $C$ is the median concentration among motile microbes inside patches, $C_P$ is the median concentration among non-motile particles inside patches (note $C_P$ is computed from the non-motile simulation, not assumed from a uniform distribution of non-motile microbes), and $C_M$ is a normalisation factor equal to the overall concentration of microbes in each simulation. $Q$ is thus dimensionless, and captures the difference in patch concentration between motile and non-motile microbes; the larger the $Q$-value, the more motile microbes are concentrated within patches than non-motile microbes. For all results reported in this paper, we chose $f = 0.01$ so that our patches consist of the 1% most-concentrated microbes. We note that our use of a reflective boundary condition at the upper boundary of the fluid simulation may not reflect the true dynamics of gyrotactic particles at the very surface of a fluid, and we therefore did not include microbes residing in the top-most DNS cell layer ($0.299\overline{166}$ m $< z \leq 0.3$ m) in the $Q$-analysis.

## Supporting information

**S1 Text. Timestepping & numerical accuracy in the microbe IBM.**
(PDF)

**S2 Text. Scaling Arguments.**
(PDF)

**S3 Text. Supplementary Figures. Fig A:** Per-Timestep Error between $\Delta t = 0.01$ s and $\Delta t = 0.001$ s. **Fig B:** Distribution of Cellwise Velocity Magnitudes across all timesteps. **Fig C:** Top-down image of microbe positions and patches at $t = 60$ s in the Deep region of the $(B, v_{\text{swim}}) = (1$ s, $500$ µm s$^{-1})$ motile simulation. **Fig D:** Sample of 2 3D microbe trajectories with superimposed microbe orientations. **Fig E:** Q statistic over time in the Shallow region of each simulation. **Fig F:** Q statistic over time in the Mid region of each simulation. **Fig G:** Q statistic over time in the Deep region of each simulation. **Fig H:** Normalised distributions of polar angle of microbe orientation in each simulation. **Fig I:** Empirical cumulative distribution functions (eCDFs) of vertical fluid velocity around microbes in the Deep region of each simulation. **Fig J:** Distribution of absolute microbe concentrations within patches for the non-motile and $(B = 1.0$s, $v_{\text{swim}} = 10$µm s$^{-1})$ simulation. **Fig K:** Distribution of 585 absolute microbe concentrations within patches for the non-motile and 586 $(B = 1.0$s, $v_{\text{swim}} = 100$µm s$^{-1})$ simulation. **Fig L:** Distribution of absolute microbe concentrations within patches for the non-motile and $(B = 1.0$s, $v_{\text{swim}} = 500$µm s$^{-1})$ simulation. **Fig M:** Distribution of absolute microbe concentrations within patches for the non-motile and $(B = 3.0$s, $v_{\text{swim}} = 10$ µm s$^{-1})$ simulation. **Fig N:** Distribution of absolute microbe concentrations within patches for the non-motile and $B = 3.0$s, $v_{\text{swim}} = 100$µm s$^{-1}$ simulation. **Fig O:** Distribution of absolute microbe concentrations within patches for the non-motile and $(B = 3.0$s, $v_{\text{swim}} = 500$µm s$^{-1})$ simulation. **Fig P:** Distribution of absolute microbe concentrations within patches for the non-motile and $(B = 5.0$s, $v_{\text{swim}} = 10$µm s$^{-1})$ simulation. **Fig Q:** Distribution of absolute microbe concentrations within patches for the non-motile and $(B = 5.0$s, $v_{\text{swim}} = 100$µm s$^{-1})$ simulation. **Fig R:** Distribution of absolute microbe concentrations within patches for the non-motile and $(B = 3.0$s, $v_{\text{swim}} = 500$µm s$^{-1})$ simulation. **Fig S:** Q-statistic over time (solid lines) in different depth regions of simulations wherein the microbes cannot reorient themselves towards the vertical (i.e. microbes with no $B$-parameter). **Fig T:** Normalised distributions of the magnitude and polar angle of effective velocity in each depth region of two simulations wherein the microbes cannot reorient themselves towards the vertical (i.e. microbes with no $B$-parameter). **Fig U:** Violin plot comparison of the distribution of $Q$-values in the truncated 0–20s simulation for agile and non-agile microbes in the combined Shallow-Mid regions and the Deep region. **Fig V:** Violin plot comparison of the distribution of $Q$-values in the truncated 0–30s simulation for agile and non-agile microbes in the combined Shallow-Mid regions and the Deep region. **Fig W:** Violin plot comparison of the distribution of $Q$-values in the truncated 0–40s simulation for agile and non-agile microbes in the combined Shallow-Mid regions and the Deep region. **Fig X:** Violin plot comparison of the distribution of $Q$-values in the truncated 0–50s simulation for agile and non-agile microbes in the combined Shallow-Mid regions and the Deep region. **Fig Y:** Boxplots of the ratio of microbial swimming speed ($v_{\text{swim}}$) to horizontal fluid velocity ($|(v_{\text{fluid},x}, v_{\text{fluid},y})|$) in the Deep region for each of the gyrotactic motile simulations.
(PDF)

## Acknowledgments

The authors would like to thank J.P. Mollicone for his assistance with the direct numerical simulation reported here.

## Author Contributions

**Conceptualization:** Alexander Kier Christensen, Matthew D. Piggott, Erik van Sebille, Samraat Pawar.

**Data curation:** Alexander Kier Christensen, Maarten van Reeuwijk.

**Formal analysis:** Alexander Kier Christensen.

**Funding acquisition:** Samraat Pawar.

**Investigation:** Alexander Kier Christensen.

**Methodology:** Alexander Kier Christensen, Matthew D. Piggott, Erik van Sebille, Maarten van Reeuwijk, Samraat Pawar.

**Project administration:** Alexander Kier Christensen, Matthew D. Piggott, Erik van Sebille, Samraat Pawar.

**Resources:** Matthew D. Piggott, Erik van Sebille, Maarten van Reeuwijk, Samraat Pawar.

**Software:** Alexander Kier Christensen, Erik van Sebille.

**Supervision:** Matthew D. Piggott, Erik van Sebille, Samraat Pawar.

**Validation:** Alexander Kier Christensen, Matthew D. Piggott, Erik van Sebille, Maarten van Reeuwijk, Samraat Pawar.

**Visualization:** Alexander Kier Christensen.

**Writing – original draft:** Alexander Kier Christensen.

**Writing – review & editing:** Alexander Kier Christensen, Matthew D. Piggott, Erik van Sebille, Maarten van Reeuwijk, Samraat Pawar.

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
