## [Decision Letter · Decision Letter 0]

22 Sep 2021

Dear Mr Christensen,

Thank you very much for submitting your manuscript "Investigating microscale patchiness of motile microbes driven by the interaction of turbulence and gyrotaxis in a 3D simulated convective mixed layer" for consideration at PLOS Computational Biology.

As with all papers reviewed by the journal, your manuscript was reviewed by members of the editorial board and by several independent reviewers. In light of the reviews (below this email), we would like to invite the resubmission of a significantly-revised version that takes into account the reviewers' comments.

We cannot make any decision about publication until we have seen the revised manuscript and your response to the reviewers' comments. Your revised manuscript is also likely to be sent to reviewers for further evaluation.

Sincerely,

Kiran Raosaheb Patil, Ph.D.

Deputy Editor

PLOS Computational Biology

Reviewer's Responses to Questions

**Comments to the Authors:**

Reviewer #1: The authors study the creation of patchiness in a microbial marine ecosystem due to the coupling of the active motility of the microorganisms and the turbulence in the water. Their conclusions confirm the hypothesis that the patchiness is due to the active swimming in the horizontal direction, but only for relatively slow velocities of the turbulent eddies in the water. The resulting patchiness occurs only in the deeper layers of the water, where the flow and the mixing are relatively low.

The topic of research and the scientific results in this paper are, I believe, of interest to a broad community of microbiologists, marine biologists, biophysicists etc. The paper is very well written, and I would like to recommend it for publication, given that the authors address the following concerns, questions and suggestions.

My main concern is about the data presented in Fig. 3. While it is clear that the value of Q for the agile microbes in the deep region shown in Fig. 3f is non-zero in average, there is a clear trend downward in the data, with a sign change near time=50. Similarly, although not as substantially as in Fig. 3f, in Figs. 3b and d, one can see a trend in the data toward negative values. One must wonder what would have happened if the time of simulation was much longer. The authors do address somewhat in the Discussion the fact that this study cannot inform us about the life-time of the patches. However, even if the life-time of the patches is on the order of time=50, the really relevant question here is the characteristic time of the changes of Q on the long run. Will Q settle to a steady non-zero value after long time, and will that value be positive or negative? Or will it settle at zero and this phenomenon of patchiness is only present in the early stages of a process that proceeds as simple mixing at longer times? In the second case, this study may have revealed the process of creation of the patches, but did not give an answer to the question of the maintenance of the patchiness. Perhaps Q oscillates between positive and negative values with the characteristic time of 50. In that case, if possible, a simulation of time=300 or so would be very illuminating, and the results for the average Q presented in the paper would not be truly relevant. I hope the authors can address this issue in the paper.

My other suggestions are of technical nature, with the purpose to make the paper easier to understand by a broad audience.

1. I think it would be very useful for the reader if the authors present an image of the patches in the deep region. Perhaps a horizontal cross section showing color coded values of Q.

2. In Fig.1 the blue color shows DeltaRho/Rho, in the text, I believe the same variable is named RhoPrime/Rho. Or perhaps DeltaRho=RhoPrime-Rho, then the caption in Fig. 1 should be changed.

3. In Fig. 7, the fonts are too small. What are DeltaY and z->x arrow?

4. In Fig. 7 I think it would be better if the units are meters, not grid points.

5. Please mention if any, and what kind of stochasticity (fluctuations) were implemented in the model.

6. I don’t believe that the distinction between active and passive swimmers is explicitly defined anywhere in the text. The reading of the text could be easier if mentioned that v_swim=0 for passive swimmers.

7. The stability and swimming numbers are defined in the Discussion. I think the readers would benefit if they are defined earlier in the Results section.

8. Units of time are needed in Fig. 3.

Reviewer #2: In this paper, Christensen et al combine individual-based simulations and direct numerical simulations of turbulent flow to investigate the emergence of patchiness in suspensions of gyrotactic microswimmers in situations mimicking phytoplankton dynamics in the upper layers of the ocean/lake water column during cooling events. Their main results are that patchiness emerges in regions where turbulence intensity is moderate to weak, situated at lower depths in their simulations, as it would be expected in a real-world water column, and only for faster cells; that high levels of turbulences homogenize the suspended microbes; this suggests that patchiness exists as a result of a balance between motility and turbulence strengths, and only at intermediate level of turbulence intensities. They also show that fast swimming, fast reorienting (“agile”) microbes are able to considerably enhance their mobility in presence of turbulence.

This work is potentially interesting to the readership of PLOS computational biology. I would however like several of my concerns to be thoroughly addressed before publication can be considered.

Major points:

Contextualisation: I think contextualization could be improved slightly. This work is inscribed in a fairly long string of related computational works on the same subject. As the author clearly acknowledge, patchiness has been shown to emerge from the interaction of gyrotactic motility and turbulence before (ref. 32, 33, 36). It should also be acknowledged that patchiness emergence has been shown and investigated in DNS of 3D turbulence before, albeit with slightly different geometries (De Lillo et al doi: 10.1103/PhysRevLett.112.044502, Marchioli et al, doi:10.1103/PhysRevFluids.4.124304). It even seems that depth and fluid forcing dependent patchiness was reported before (doi:10.3354/meps12490), albeit with very different forcing and simulation techniques.

Impact: Given the above mentioned context, simply concluding that patchiness emerges in a given regime of turbulence/motility relative strength in the geometry used in the paper gives the impression of a fairly limited impact, even considering that the simulations indeed closer mimick real world conditions. To raise impact, I think the authors should investigate more thoroughly the physical mechanisms that lead to the behavior, in particular test the actual impact of coupling of fluid shear and motility, as was suggested they would do in the paragraph starting l45 in the introduction. I think it is quite feasible given the extent and quality of the simulation data at hand. In particular:

- How important is motility to patchiness emergence, how (in)homogeneous is a population of non-motile cells in the same turbulent flow? Q characterizes excess patchiness relative to non-swimmer case. It would be interesting to see C/C_M drawn separately for simulations of motile and non-motile to have a better idea of absolute levels of heterogeneities even for non-swimmers in the turbulent flows. Other metrics such as the analysis of variance of the mean density in a box as a function of box size, commonly used in statistical physics to characterize giant fluctuations, or fractal analysis (e.g. De Lillo et al, doi:10.1103/PhysRevLett.112.044502) could be used to compare the swimmer and non-swimmer cases, if C/C_M proves impractical.

- Related to this: Are non-agile cells for all instance and purposes similar to non-motile cells in these simulations? i.e. What are the distributions of orientation and total speed (Fig 5 and 6) for non-motile cells (either gyrotactic or not)?

- How important is directionality of gyrotaxis to the emergence of patchiness? From equation (3), Brownian rotational motion is neglected. What happens if the directed gyrotactic reorientation process is replaced by an isotropic Brownian reorientation process of the same speed? Is patchiness still observed at v =500 um/s?

- Relatedly: in Fig 5 + associated SI7: Slow reorienting cells have a preferred orientation at the horizontal throughout the water column. I find this very strange considering that the cells are round, and should have homogenized orientations if gyrotaxis is totally ineffective. Is it also the case in non-gyrotactic/passive case? How can this horizontal preference be explained?

- Only “agile cells” (v=500 um/s, B=1 or 3 s) are both more patchy than non-motile ones and more advected than the non-agile ones. How do these two aspects relate to each other? Are the fastest agile cells at any given time also the ones in denser (/looser) patches? In other words, how do structure and dynamics correlate?

- The authors make an attempt in the discussion with Fig S8 at explaining the enhanced mobility of agile cells, but I find their explanation slightly tautological, since it amounts to these cells being advected faster by the flow, which was expected since Phi is lower and even much lower than 1 in all tested conditions. The interesting question in my opinion is: How/why do they migrate into the fastest parts of the flow? Interaction between sheared flows and swimming has been extensively studied. Some active swimmers are known to be expelled from regions of high shear, as noted in the introduction (see also Sokolov & Aronson doi: 10.1038/ncomms11114). Can it be shown to be the case here, and if so, does that explain densification and enhanced transport? (What is the distribution of shear rates (and vorticity) experienced by the agile swimmers? How does it compare to non-agile case? Are shear rate (vorticity) and advection velocity anti-correlated?)

Technical points:

- In Figure 3 and Fig S4-S6, it seems that either the dynamics of the microbes during the IBM do not reach a steady state in the more “agile” cases or that only a small fraction of the phase space is sampled. Longer simulations should be conducted to show that the system reaches a steady state in these cases.

- The authors carefully characterized their numerical errors in SI. However, since interesting phenomena only appear in the “deep” region, it would be useful as an additional control to show the spatial distribution of the instances of high numerical errors, in particular to show that they are not enriched there.

Minor points:

Please introduce the definition and physical meaning of Psi and Phi in the introduction, as general readership will not know what they are.

Fig. 2 Please modify the figure to explicitly show that the green ‘less/more patchy’ arrows are expectations and not results.

Fig. 5 Parameter B should be in s, not in s^-1

L 148 please refer to SI Fig S7 to support this statement…

L 339 please rename the buoyancy flux to another letter than B which is also the gyrotactic reorientation time.

Control simulations with passive objects: it is not clear from reading the methods whether such simulations were actually conducted. Fig S8 and Author summary suggests that yes, but it should be explicitly specified in the paragraph describing the IBM, as well as the equation of motion of the passive particles

Relatedly: L 438 Please specify if Cp is calculated e.g. assuming random distribution or derived from simulations

Reviewer #3: Reviewing Christensen et al.: "Investigating microscale patchiness of motile microbes driven by the interaction of turbulence and gyrotaxis in a 3d simulated convective mixed layer".

The authors incorporate an individual-based model of microbial behavior into a 3D DNS of field-relevant convective mixed layer turbulence and demonstrate a depth-dependent patchiness occurs, but that it relies heavily on strong swimming and weak turbulence in the lower part of the mixed layer. I think that a modelling study investigating patchiness in this kind of realistic depth-varying anisotropic turbulence is useful and relevant to the community, and it is grounded with an extremely clear and well written introduction and motivation. I have some questions about the validity of some parameter choices that could be better justified, along with some questions about the presentation of the results, that would improve manuscript clarity and readability, as given below.

Main comments:

1. The paper is structured a bit oddly with respect when and where parameters are defined, which hinders understanding and interpretation of the results.

a) For instance, the dimensionless parameters phi and psi (p.6, lines 106-107) are brought up in the results but (oddly) not defined until the discussion (p.11, lines 183-184), and don't appear to be presented in the main methods text at all --though perhaps I missed them? These parameters ought to show up somewhere in the introduction, or at least briefly defined at the beginning of the results, as they are necessary to understand the results. It also makes Figure 2 quite hard to interpret --one is not sure how these parameters emerge from the various model runs and and as such it is hard to get any particular message from Figure 2.

b) Likewise, it is not clear from the introduction/results how to interpret Q, which is qualitatively defined in the results but does not receive a full treatment until the methods. However, when one reads the paper in order, statements like "the larger the Q value the more motile microbes were concentrated..." (p.6, lines 103-104) do not really give sufficient information. Particularly, there is insufficient grounding given to understand what it means for Q to be positive or negative, as the mean is in several panels of Figure 3. The results should give sufficient detail for this interpretation for a reader going through the paper in order.

2. Some additional framing is necessary throughout the manuscript to justify the choices of parameters for the microbial swimming states ('agile' vs 'non-agile'). These are presented in the results (p.6, lines 109-112) describing 'fast' swim speeds of 100-500um/s and 'slow/intermediate' swim speeds of 10-100um/s. Leaving aside the question of reorientation times, this so-called agile microbe has a very wide range of swimming speeds, and much of this range (I believe) lies outside realistic swimming velocities for gyrotactic cells. Chlamydomonas is one of the speediest gyrotactic cells I know of, and even that maxes out under 200um/s; the references [69-71] provided by the authors as justification also give these <200um/s concrete values. While Jones et al. (1994) as cited by the authors gives an upper bound of 500um/s for a generic swimming cell (Table 1), this lacks any reference to a known species. For this reason, because the authors use 'agile' to describe such a wide array of possible speeds (at least in the main text), it is hard to tell how many of their results are widely applicable. Of course, as a modelling study, some latitude in parameter selection is of course acceptable but the authors can do more (particularly in the Discussion) to frame their results in an ecological context.

3. The results of Figure 6 are very counterintuitive to me; why would a vertically-swimming, bottom-heavy gyrotactic cell swim horizontally, (most strongly) in the deep layer where the flow is most quiescent? A naive assumption would be that the cell swimming would act more strongly in the deep layer than in the more turbulent surface and mid layers. An explanation is necessary here, particularly because the Kolmogorov timescale in the deep section is ~10s (p.13, line 201) and the agile microbes have a reorientation timescale of 1-3s (p.6, line 110), which to me suggests that swimming ought to overpower flow constraints. Some clarification on this point would be very appreciated.

Minor comments:

-Figure 1: Text uses rho' (p.5, line 88) while the figure colorbar label uses delta rho. This should have a single notational convention.

-Figure 1: it is more conventional from an oceanographic perspective to set z=0 at the surface rather than the deepest part of the simulated depth; I would recommend reversing the y-axis conventions here and in the text.

-p.5, lines 94-98: Were the depth regions qualitatively selected to divide the simulation space according to distinct flow parameters, or was there a systematic selection process to divide the space? A clarifying sentence here would help.

-Figure 3: How do you interpret the trend towards negative Q statistic values over time, particularly in the agile swimmers?

-Figure 3: A minor stylistic point, but the panels are labelled twice: once in the top left corner in bolded text, and once in the bottom left corner --this latter set of labels should be removed.

-Figure 5: Are these distributions of microbe orientations averaged over the full depth of the simulation? This should be made clear in the text.

-p.11, lines 170-171: A nitpick, but since authors aren't conducting a statistical analysis the use of 'significantly greater' is misleading. This should be rephrased.

-p.15-16, lines 282-295: While it is very interesting that motile cells can boost their effective velocity by encountering fast-moving packets of fluid, it would help to have a clear idea of where these 'expressways' carry cells. Do they purely transport microbes upward or downward in the water column in convective cells?

-Figure 7: Why aren't the axes reported in dimensional units rather than DNS grid units? Likewise the velocities should be reported in um/s to make it more readily comparable to the swimming speeds of the microbes. Finally, a demarcation in (a) showing where the shallow, mid, and deep regions lie (as shown in Figure 1) would be helpful here.

**Have the authors made all data and (if applicable) computational code underlying the findings in their manuscript fully available?**

Reviewer #1: Yes

Reviewer #2: **No: **Codes have been taken from previous work. I did not find an explicit mention that no modifications were implemented from the original codes. Please clarify. Moreover, the data folder provided at https://doi.org/10.17605/OSF.IO/72YNH turned out to be empty when I tried to download it. I presume something have gone wrong here.

Reviewer #3: None

PLOS authors have the option to publish the peer review history of their article (what does this mean?). If published, this will include your full peer review and any attached files.

Reviewer #1: No

Reviewer #2: No

Reviewer #3: No
---

## [Decision Letter · Decision Letter 1]

20 Apr 2022

Dear Dr Christensen,

Thank you very much for submitting your manuscript "Investigating microscale patchiness of motile microbes under turbulence in a simulated convective mixed layer" for consideration at PLOS Computational Biology. As with all papers reviewed by the journal, your manuscript was reviewed by members of the editorial board and by several independent reviewers. The reviewers appreciated the attention to an important topic. Based on the reviews, we are likely to accept this manuscript for publication, providing that you modify the manuscript according to the review recommendations.

Sincerely,

Kiran Raosaheb Patil, Ph.D.

Deputy Editor

PLOS Computational Biology

[LINK]

Reviewer's Responses to Questions

**Comments to the Authors:**

Reviewer #1: The authors have addresses my questions, concerns and comments. I would like to recommend the paper for publication. However, the link for your public data https://osf.io/72yn does not seem to be working. Please make sure that it is accessible.

Reviewer #2: The authors addressed most of my comments. I notably appreciate the clarifications to the text, and the simulations clarifying the essential role of reorientation in the patchiness enhancement. I believe that Suppl Fig 20 and 21 could be moved to the main text, as they emphasize the essential role of a reorientation mechanism to enhance both patchiness and mobility in the deep region. I would have preferred to see longer simulations for figure 3, but the authors instead now discuss clearly this limitation of their simulations, which is totally fair in my opinion. Before acceptance, I would however want to raise a couple additional minor points:

1. Fig. SI 10-19: The figures did not seem to display properly on my pdf, the axis bars being moved to random places. The legends are not labeled properly: the value of B is different from the one indicated in the figure for graphs 14-19. It is not clear whether this is the distribution of large densities across the whole water column or only for the deeper region. The time frame over which the distribution is sampled is also not indicated.

More worryingly, it also shows that the differences between the patch size distributions in motile and non-motile cells are very subtle to say the least. In particular suppl Fig 13 and 16, which seems to be the genuine B=1 s & B=3s, v =500 µm/s simulation, exhibits barely any difference between motile and non-motile cells – if anything the distribution of non-motile density patches is wider, despite being the ones for which Q>0 in Fig 3… It could be that this is due to the distribution being calculated across the whole water column; considering only the deeper region, across the time range where Q peaks, would be probably a better choice in this case.

Related to this, I now think that a statistical analysis should be performed to demonstrate that the Q>0 excursion of Fig. 3f is statistically significant.

2. Figure 5: the legend still indicates B in s-1 instead of s.

3. In suppl figure 20 and 21, and related text in the main text, no reorientation corresponds to B->\\infty or 1/B=0, but not B=0 as was written a couple of time.

4. Although I appreciate the 1/B=0 simulation, I partially disagree with the authors on their suggestion that simulations with Brownian motion do not bring more information: In simulations with no-reorientation, swimming cells have no way to escape alignment and thus confinement within the streamlines via the (w x p) term of eq 3; in the gyrotactic case, this escape is provided by an *anisotropic* reorientation mechanism, whereas the Brownian case provides an escape by an *isotropic* reorientation mechanism. This is interesting from a physical perspective, to understand the role of the anisotropy, but also from a biological perspective because bacterial microbes, which are an important part of the oceanic ecosystems, rarely show gyrotaxis, contrary to microalgae, but do have isotropic reorientation mechanisms (tumbles, flicks and Brownian rotational motion). Of course I would love to see simulations with a Brownian reorientation mechanism, but I realize that this is quite a big endeavor, that might fall out the scope of the paper. But this point should nonetheless be discussed.

If these minor points are addressed, I believe that the paper may be accepted in PloS Computational Biology.

Reviewer #3: No further comments.

**Have the authors made all data and (if applicable) computational code underlying the findings in their manuscript fully available?**

Reviewer #1: **No: **The link https://osf.io/72yn they listed does not work. Needs to be fixed. The other links are fine, the code is publicly available.

Reviewer #2: None

Reviewer #3: None

PLOS authors have the option to publish the peer review history of their article (what does this mean?). If published, this will include your full peer review and any attached files.

Reviewer #1: No

Reviewer #2: No

Reviewer #3: No

Figure Files:

Data Requirements:

Reproducibility:

References:

---

## [Editor Report · Decision Letter 2]

9 Jun 2022

Dear Dr Christensen,

We are pleased to inform you that your manuscript 'Investigating microscale patchiness of motile microbes under turbulence in a simulated convective mixed layer' has been provisionally accepted for publication in PLOS Computational Biology.

Best regards,

Kiran Raosaheb Patil, Ph.D.

Deputy Editor

PLOS Computational Biology

---

## [Editor Report · Acceptance letter]

22 Jul 2022

PCOMPBIOL-D-21-01230R2 

Investigating microscale patchiness of motile microbes under turbulence in a simulated convective mixed layer

Dear Dr Christensen,

I am pleased to inform you that your manuscript has been formally accepted for publication in PLOS Computational Biology. Your manuscript is now with our production department and you will be notified of the publication date in due course.

With kind regards,

Agnes Pap
